# Climate warming reduces the temporal stability of plant community biomass production

Zhiyuan Ma[1,*], Huiying Liu[1,*], Zhaorong Mi[2,†], Zhenhua Zhang[2], Yonghui Wang[1,†], Wei Xu[1], Lin Jiang[3] & Jin-Sheng He[1,2]

Anthropogenic climate change has emerged as a critical environmental problem, prompting frequent investigations into its consequences for various ecological systems. Few studies, however, have explored the effect of climate change on ecological stability and the underlying mechanisms. We conduct a field experiment to assess the influence of warming and altered precipitation on the temporal stability of plant community biomass in an alpine grassland located on the Tibetan Plateau. We find that whereas precipitation alteration does not influence biomass temporal stability, warming lowers stability through reducing the degree of species asynchrony. Importantly, biomass temporal stability is not influenced by plant species diversity, but is largely determined by the temporal stability of dominant species and asynchronous population dynamics among the coexisting species. Our findings suggest that ongoing and future climate change may alter stability properties of ecological communities, potentially hindering their ability to provide ecosystem services for humanity.

[1] Department of Ecology, College of Urban and Environmental Sciences, and Key Laboratory for Earth Surface Processes of the Ministry of Education, Peking University, Beijing 100871, China. [2] Key Laboratory of Adaptation and Evolution of Plateau Biota, Northwest Institute of Plateau Biology, Chinese Academy of Sciences, Xining 810008, China. [3] School of Biological Sciences, Georgia Institute of Technology, Atlanta, Georgia 30332, USA. * These authors contributed equally to this work. † Present addresses: Farmland Irrigation Research Institute, Chinese Academy of Agricultural Sciences, Xinxiang 453002, China (Z.M.); Department of Ecology, College of Life Sciences, Inner Mongolia University, Hohhot 010021, China (Y.W.). Correspondence and requests for materials should be addressed to L.J. (email: lin.jiang@biology.gatech.edu) or to J.-S.H. (email: jshe@pku.edu.cn).

One of the basic attributes of an ecological system is its stability[1,2]. Since stable ecosystems are important for providing sustainable ecosystem functioning and services to humanity[3], understanding the drivers of ecological stability has emerged as a pressing issue in a period when many ecosystems are experiencing significant anthropogenic change[4–7]. Much recent attention has focused on the temporal stability of community biomass production, known to be influenced by several mutually nonexclusive mechanisms. First, changes in biodiversity may result in changes in biomass temporal stability[6–8]. On the one hand, biodiversity may influence biomass temporal stability because more diverse communities are more likely to contain species that are resistant to environmental fluctuations (the sampling effect[9–12]). On the other hand, biodiversity can have a positive influence on biomass stability via promoting asynchronous population dynamics among species[13–15]. Second, absent biodiversity change, variation in the degree of asynchrony in population dynamics, in response to variation in abiotic and biotic conditions, may also lead to changes in community biomass stability[6,16,17]. Third, the population stability of dominant species may strongly influence community biomass stability, especially when communities are dominated by a small number of species[18–20]. Hence, climate change that alters biodiversity, the degree of species asynchrony and/or the stability of dominant species may have the potential to alter the temporal stability of community biomass production.

The average global temperature has increased by 0.065 °C per decade since 1880, accompanied by significant changes in precipitation patterns[21]. These rapid climate changes, unprecedented in human history, are likely to profoundly affect the functioning of Earth's ecosystems[22,23]. Climate warming has been demonstrated to influence community structure[24–26] and species interactions[27–29] that could potentially translate into changes in community biomass stability[30–32]. Likewise, changes in precipitation could alter community biomass production[33], species diversity[34] and species relative abundance patterns[35] that may also have consequences for biomass stability[16,17]. Notably, the effects of climate warming and altered precipitation on community properties, including stability, may not be independent of each other, given that warming is expected to impose its strongest effect under drought conditions[33,36,37]. Studies manipulating both warming and precipitation to assess their influence on community biomass stability are vital for predicting ecosystem dynamics under future climate change scenarios.

Here, we report on a 5-year field experiment investigating the influence of climate warming and altered precipitation on the temporal stability of plant biomass production of an alpine grassland on the Tibetan Plateau (Supplementary Fig. 1). The Tibetan Plateau has an area of 2.5 million km[2], with 64% of this region occupied by alpine grassland that provides essential ecosystem services for humans living in the region[38]. As the world's highest plateau (4,500 m above sea level on average), the Tibetan Plateau has experienced more rapid climate warming (0.4 °C per decade over the past 50 years) than average, coupled with increasing and greater interannual variation in precipitation[38,39]. Climate warming and altered precipitation are known to influence net primary production[24], biogeochemical cycles[38], litter decomposition[40] and rangeland quality[41] in this region. However, their effects on the temporal stability of this important ecosystem are unknown. We aimed to explore how future climate change would affect biomass temporal stability and elucidate the role of potential mechanisms in driving the observed stability response to climate change. We show that climate warming reduces the temporal stability of community biomass via reducing the degree of species asynchrony, independent of

precipitation effects. This result suggests that ongoing and future climate change may reduce the ability of the alpine grassland and other similar ecosystems to provide reliable ecosystem services for humanity.

## Results

**Plant community response to climate change.** Over the 5-year experimental period, warming did not influence plant community biomass (linear mixed-effects model: $P = 0.84$; Fig. 1a), but increased precipitation resulted in an average of 17.5% increase in

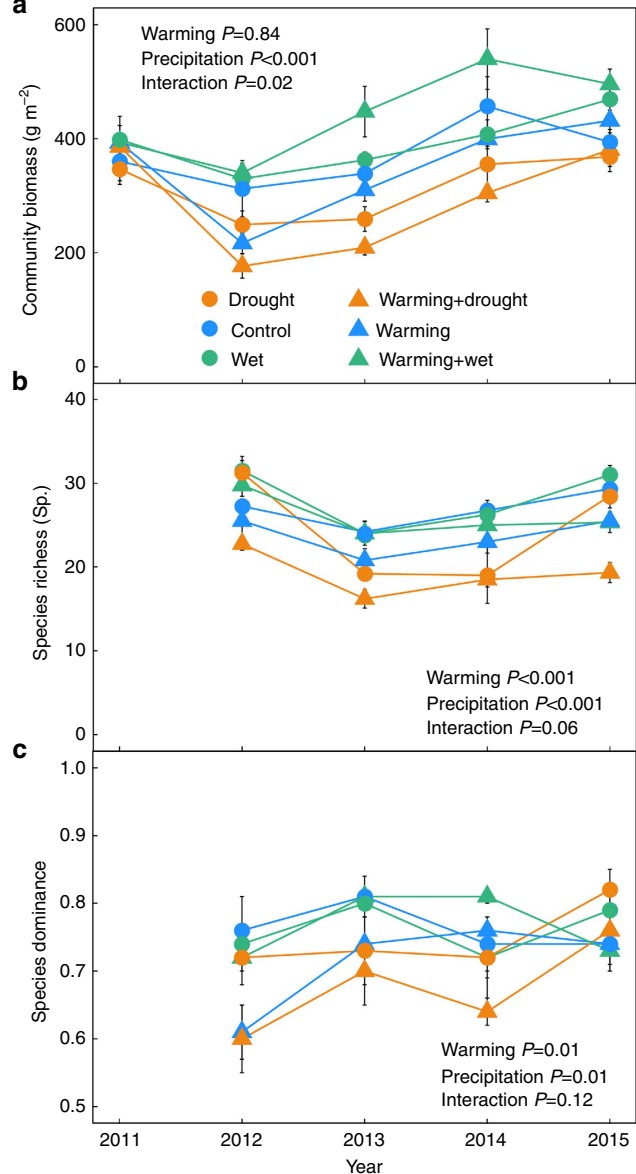

**Figure 1 | Community biomass and species diversity in different experimental treatments.** Shown are (**a**) community biomass (linear mixed-effects model; warming: $P = 0.84$; precipitation: $P < 0.001$; interaction: $P = 0.02$); (**b**) species richness (linear mixed-effects model; warming: $P < 0.001$; precipitation: $P < 0.001$; interaction: $P = 0.06$); and (**c**) Simpson's dominance (linear mixed-effects model; warming: $P = 0.01$; precipitation: $P = 0.01$; interaction: $P = 0.12$) in the warming and altered precipitation treatments during the experimental period. Drought, 50% reduction in precipitation compared with control; Wet, 50% increase in precipitation compared with control. Vertical bars represent the s.e.m. ($n = 5$).

plant community biomass ($P < 0.001$; Fig. 1a). The effect of altered precipitation on community biomass, however, was stronger under warming, resulting in a significant warming × precipitation interaction term in the linear mixed-effects model ($P = 0.02$; Fig. 1a and Supplementary Table 1). Both warming and increased precipitation had negative effects (a reduction of 4.9% and 14.1%, respectively) on species richness and positive effects (an increase of 3.5% and 6.6%, respectively) on species dominance (all $P < 0.05$; Fig. 1b,c); their effects were largely independent of each other (Fig. 1b,c and Supplementary Table 1).

**Community temporal stability and species asynchrony.** Warming significantly reduced the temporal stability of plant community biomass by an average of 23.2% (linear mixed-effects model: $P = 0.04$; Fig. 2a), whereas precipitation alteration did not affect community biomass stability ($P = 0.17$; Fig. 2a and Supplementary Table 1). Likewise, warming significantly decreased species asynchrony by an average of 11.9% ($P = 0.01$; Fig. 2b), whereas precipitation alteration had little effect on species asynchrony ($P = 0.37$; Fig. 2b and Supplementary Table 1). No significant interactive treatment effects were observed for either biomass temporal stability or species asynchrony.

**Dominant species stability.** Neither warming (linear mixed-effects model: $P = 0.23$) nor precipitation alteration ($P = 0.27$) affected dominant species stability (Fig. 2c). Species relative abundance was significantly positively correlated with their temporal stability in all six treatments (linear regressions: all $R^2 > 0.18$, $P < 0.01$; Supplementary Fig. 2a), indicating that dominant species were more temporally stable than less abundant species. Neither warming nor altered precipitation affected the relationship between species relative abundance and stability (analysis of covariance (ANCOVA) species relative abundance × treatment interaction: $P = 0.26$, $n = 1,079$; Supplementary Fig. 2a). Similar results were obtained when rare species, whose high population variability (that is, low stability) may be potentially inflated by their low detectability, were excluded from the analysis (all $R^2 > 0.13$, $P < 0.01$; ANCOVA species relative abundance × treatment interaction: $P = 0.85$, $n = 422$; Supplementary Fig. 2b).

**Functional group stability.** When we classified species into functional groups (grasses, sedges, legumes and forbs), we found a change in the abundance of each group in response to experimental treatments (Supplementary Fig. 3). However, the temporal stability of each functional group was unaffected by experimental treatments (linear mixed-effects models: all $P > 0.05$; Supplementary Table 2).

**Ecological factors influencing biomass temporal stability.** Linear regression revealed that community biomass temporal stability was significantly positively correlated with species asynchrony ($R^2 = 0.53$, $P < 0.001$; Fig. 3a), species richness ($R^2 = 0.15$, $P = 0.03$; Fig. 3b) and dominant species stability ($R^2 = 0.58$, $P < 0.001$; Fig. 3c), but not related to species dominance ($R^2 = 0.04$, $P = 0.30$; Fig. 3d). Among the four functional groups, only the stability of the grass functional group was significantly, positively, associated with community biomass stability (linear regression: $R^2 = 0.29$, $P < 0.01$; Supplementary Fig. 4). Structural equation modelling (SEM) showed that dominant species stability and species asynchrony jointly explained 75% of the variation in community biomass stability, and that warming, but not precipitation alteration, affected biomass temporal stability (Fig. 4 and Supplementary Fig. 5). The negative effect of warming on biomass temporal stability was mainly through its negative effect on species asynchrony

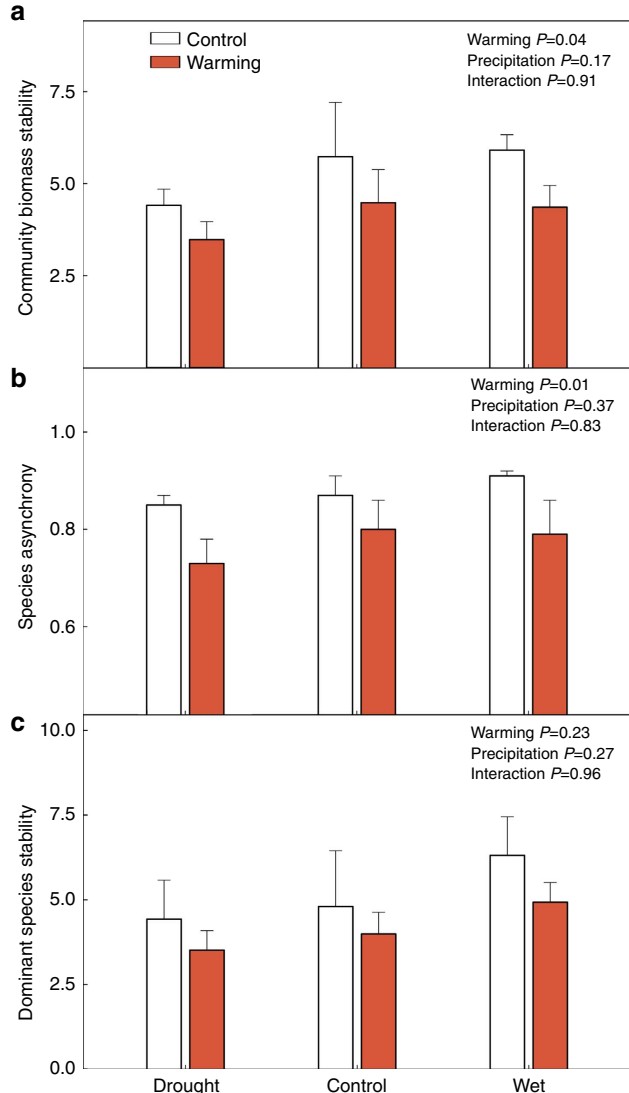

**Figure 2 | Temporal stability and species asynchrony in different experimental treatments.** Shown are (**a**) community biomass temporal stability (linear mixed-effects model; warming: $P = 0.04$; precipitation: $P = 0.17$; interaction: $P = 0.91$); (**b**) species asynchrony (linear mixed-effects model; warming: $P = 0.01$; precipitation: $P = 0.37$; interaction: $P = 0.83$); and (**c**) dominant species stability (linear mixed-effects model; warming: $P = 0.23$; precipitation: $P = 0.27$; interaction: $P = 0.96$) in the warming and altered precipitation treatments during the experimental period. Drought, 50% reduction in precipitation compared with control; Wet, 50% increase in precipitation compared with control. Vertical bars represent the s.e.m. ($n = 5$).

(Fig. 4 and Supplementary Fig. 5). Note that species richness, which showed positive association with biomass temporal stability (Fig. 3b) and species asynchrony (Supplementary Fig. 6c) in regression analyses, was eliminated from the SEM as a significant predictor of biomass temporal stability (Fig. 4, Supplementary Table 3 and Supplementary Fig. 5). Considering functional group stability in the SEM did not significantly improve the fit of the model ($R^2 = 0.77$ and 0.75 in the SEM model with and without considering grass stability, respectively).

## Discussion
Much of the Earth is experiencing climate warming and change in precipitation patterns. Ecologists have just begun to explore their

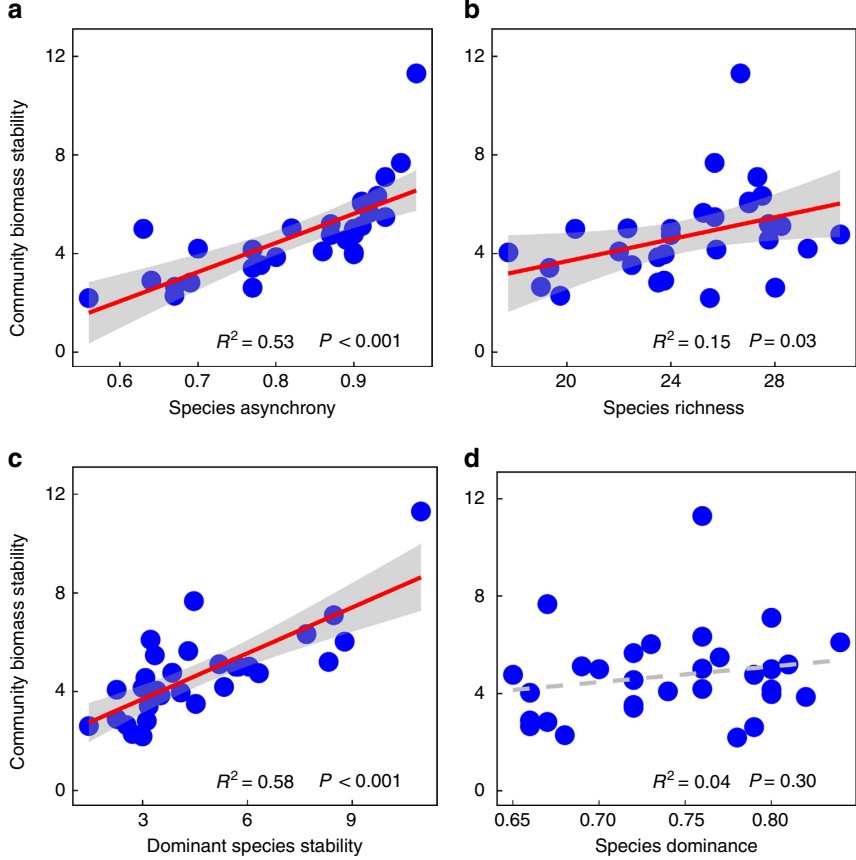

**Figure 3 | Relationships between ecological factors and community biomass stability.** Shown are (**a**) species asynchrony (linear regression; $P < 0.001$); (**b**) species richness (linear regression; $P = 0.03$); (**c**) dominant species temporal stability (linear regression; $P < 0.001$); and (**d**) Simpson's dominance (linear regression; $P = 0.30$). The red solid lines are significant regression lines, and the grey dashed lines are nonsignificant regression lines. Each blue circle represents an experimental plot ($n = 30$). Shaded areas represent 95% confidence intervals.

consequences for ecological stability; three aspects of our study, however, distinguish it from the few previous studies on this topic. First, while the few studies of climate change effects on temporal stability have considered either warming[31,32] or precipitation[16,17], our study examines both warming and precipitation effects in a single experiment. Second, our study was conducted on the Tibetan Plateau that, as the world's largest and highest plateau, is considered highly susceptible to climate change. Our study thus represents a rare exploration of climate change impacts on stability in an ecologically important but vulnerable habitat. Third and most important, our study demonstrated that climate warming could reduce biomass temporal stability via reducing species asynchrony, independent of precipitation effects.

Many studies, including field observations[42,43], experiments that directly manipulated diversity[8,44–46] and theoretical models[47–49], have shown that increasing species diversity tends to increase biomass temporal stability. In our study, we also found a significant positive relationship between species diversity and biomass temporal stability as revealed by linear regression (Fig. 3b). Given that climate warming reduced species diversity in our experiment, as also found in other studies conducted in the same ecosystem[24,50], one might think warming-induced diversity decline contributed to the decline in biomass stability under warming. However, species diversity was not retained in the SEM as a significant predictor of biomass stability (Fig. 4). On the other hand, increasing precipitation significantly increased species diversity, but precipitation alteration did not affect biomass stability (Figs 1 and 4 and Supplementary Fig. 5).

One possible explanation for the lack of species diversity effect on biomass stability in our experiment is that changes in species richness (2–4 species) under our experimental treatments were relatively small compared with community-level species richness (~26 Sp.). Probably more importantly, most of the lost/gained species in response to our experimental treatments were rare species that accounted for only a small fraction of community biomass (Supplementary Table 4). These rare species contributed relatively little to community biomass stability compared with species asynchrony and the stability of dominant species (see below), making species diversity an unimportant driver of community biomass stability.

Asynchrony in population dynamics across species is a common feature of ecological communities[51,52] and could be attributed to asynchronous species responses to environmental fluctuations[2,15]. The most important finding of our study is that climate warming reduced species asynchrony, translating into reduced community biomass stability under warming. Under climate warming, a few species, including *Helictotrichon tibeticum* and *Medicago archiducis-nicolai*, increased in abundance. However, the majority of species, which constituted 75% of community members, declined in abundance under warming, resulting in less asynchronous population dynamics. This result contrasts with that of a warming experiment in a temperate steppe, where neither daytime nor nighttime warming altered species asynchrony[32]. This discrepancy may be explained by temperature being a stronger limiting factor (that is, cold temperature constrains biological processes[29]) in the alpine grassland than the temperate steppe, translating into stronger

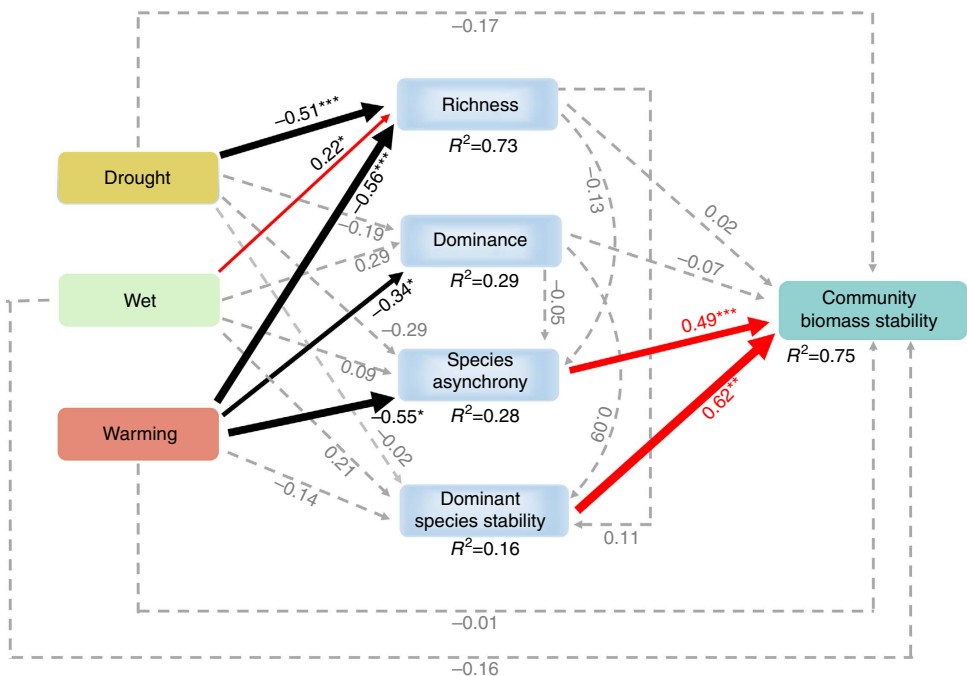

**Figure 4 | A structural equation model of treatment effects on biomass temporal stability.** The structural equation model considered all plausible pathways through which experimental treatments influence community biomass temporal stability. Red and black arrows represent significant positive and negative pathways, respectively, and grey dashed arrows indicate nonsignificant pathways. Bold numbers indicate the standard path coefficients. Arrow width is proportional to the strength of the relationship. $R^2$ represent the proportion of variance explained for each dependent variable in the model. *$P < 0.05$, **$P < 0.01$, ***$P < 0.001$; $\chi^2 = 7.27$, $P = 0.03$; root mean square error of approximation (RMSEA) = 0.30, $P = 0.03$; Akaike information criteria (AIC) = 334.90.

species responses to the same amount of warming in the alpine grassland. Probably also because of the limitation of plant growth by cold temperatures, we found that precipitation alteration did not influence species asynchrony or biomass stability. This contrasts with the results of several studies conducted in temperature grasslands, where increased precipitation was found to promote species asynchrony[16,17]. Together, these results highlight the context dependency of how climate changes influence biomass stability and associated mechanisms, emphasizing the need to explore the combined effects of warming and altered precipitation in other ecosystems.

A growing body of evidence suggests that community biomass stability tends to show positive associations with the stability of dominant species[5,17,19,53–56] that may even constrain the effects of species diversity on biomass stability[57]. Our findings reinforce these ideas. An important characteristic of dominant species in our study is their greater temporal stability than less abundant species (Supplementary Fig. 2), a finding that has also been reported by several other studies[32,54,57]. Importantly, neither climate warming nor altered precipitation affected the temporal stability of these dominant species that exhibited strong positive relationships with community biomass stability. In our experiment, the two dominant species, *Stipa aliena* and *Elymus nutans*, accounted for approximately half (48.7%) of community biomass. They were relatively insensitive to environment fluctuations, probably because of their greater nutrient acquisition ability through well-developed root systems, and their greater light acquisition ability through taller canopy and larger specific leaf area in this alpine region[24]. Common and rare species represented 30.2% and 66.0% of species richness, respectively, but they accounted for only 40.1% and 11.2% of community biomass, respectively (Supplementary Table 4). As a result, there was no significant relationship between community

temporal stability and common or rare species stability in our experiment (Supplementary Fig. 7). Overall, the significant positive relationship between dominant species stability and community biomass stability supports the mass ratio hypothesis[58], highlighting the importance of dominant species for ecosystem functioning.

Our study provides new empirical evidence that climate warming can have a negative effect on the temporal stability of community biomass production, independent of precipitation scenarios. This result suggests that future climate change may reduce the ability of our study system, the alpine grassland that covers much of the Tibetan Plateau, to provide reliable ecosystem services for humanity. Future studies should assess the generality of this result to other ecosystems. Furthermore, our study identified weakened species asynchrony as the main reason of why warming reduced the temporal stability of community biomass. Elucidating how population dynamics of individual species respond to climate change in a community context thus holds the key to understanding changes in community stability properties under future climate scenarios.

## Methods

**Study site.** Our study site is located at the Haibei Alpine Grassland Ecosystem Research Station (101°12′E, 37°37′N, and 3,200 m above sea level) on the northeastern Tibetan Plateau in Qinghai Province, China. This area has a continental monsoon climate, with a 6-month growing season (from mid-April to mid-October). The mean annual temperature is − 1.1 °C, with an extreme maximum air temperature of 27.6 °C in July, and an extreme minimum temperature of − 37.1 °C in January. The mean annual precipitation is 485 mm, with >80% of the precipitation falling in the growing season. The soil is classified as Mat-Gryic Cambisol, with the average pH value of surface soil (0–10 cm) being 6.4 (refs 59,60). The alpine grassland is dominated by perennial plants, including *S. aliena*, *E. nutans*, *H. tibeticum*, *Kobresia capillifolia*, *Carex przewalskii*, *Poa annua*, *M. archiducis-nicolai*, *Tibetia ruthenia*, *Gentiana straminea* and *Saussurea superba*. These species together account for ~70% of aboveground net primary production (g m$^{-2}$).

**Experimental design.** Our experimental plots were established within an area of 50 m × 110 m in 2011, using a randomized block design with warming and altered precipitation as main treatment factors (Supplementary Fig. 1a). Each block contained six treatments, crossing two levels of warming (no warming, warming) and three levels of precipitation (no precipitation, drought (50% precipitation reduction) and wet (50% precipitation addition)). Each treatment had five replicates, resulting in 30 plots; each plot was 1.8 m × 2.2 m. In the warming treatments, two infrared heaters (1,000 mm length, 22 mm width) were suspended in parallel at 150 cm above the ground within each plot (Supplementary Fig. 1b), with an electrical power output of 1,200 W for each heater; the heaters resulted in an increase of 2 °C above ambient temperature at the top 5 cm layer of the soil[61]. Rain shelters were used to control the incoming precipitation amount in the experimental plots. Four 'V'-shaped transparent polycarbonate resin channels (Teijin Chemical, Japan) were fixed at the 15° angle, above the infrared heaters, to intercept rainfall. The collected rainfall from the drought plots was supplied to the wet plots manually after each precipitation event by spraying bottle (Supplementary Fig. 1c). To account for the effects of shading, we also installed two 'dummy' infrared heaters and four 'dummy' transparent polycarbonate resin channels in the control plots. Stainless steel sheets were inserted into the soil around the edge of each plot to reduce surface runoff.

**Plant community monitoring.** To estimate the biomass of community and individual species, three 0.15 m × 0.15 m quadrats were randomly chosen within each plot, and clipped at the ground level in late August (the peak of growing season) from 2011 to 2015. Plants clipped from the three quadrats of each plot were pooled together, sorted to species (from 2012 to 2015), and oven-dried at 65 °C for 48 h. Plants were classified into three different groups (dominant, common and rare species) according to their relative abundance and four functional groups (grasses, sedges, legumes and forbs) based on their functional forms. Dominant species included those with relative abundance >5%, common species ranged from 1 to 5% in relative abundance and rare species were those with relative abundance <1% (refs 62–64). The three groups consisted of 2, 16 and 35 species, and accounted for 52.3%, 35.9% and 11.9% of community biomass, respectively.

**Statistical analysis.** We quantified temporal stability of community biomass as the ratio of mean biomass ($\mu$) to its temporal s.d. ($\sigma$) in each plot over the 5 years of the experiment (2011–2015), as in many other studies[6,65]; the temporal stability of individual species and functional groups were calculated using the same method over the four years when species-level data were available (2012 to 2015). Species richness in each plot was defined as the total number of species detected in the three quadrats. We also calculated Simpson's dominance index[66] based on species biomass data. The degree of species asynchrony was quantified by the community-wide asynchrony index[15], defined as:

$$1 - \varphi_x = 1 - \sigma^2 / \left( \sum_{i=1}^{S} \sigma_i \right)^2 \qquad (1)$$

where $\varphi_x$ is species synchrony, $\sigma^2$ is the variance of community biomass and $\sigma_i$ is the s.d. of biomass of species $i$ in a plot with $S$ species. This index attains one when species fluctuations are perfectly asynchronized, and attains zero when species fluctuations are perfectly synchronized.

No significant temporal trend in community biomass was detected during the experimental period; thus, no detrending was conducted. Linear mixed-effects models were used to assess the effects of warming, precipitation, year and their interactions on community/functional group biomass, species richness and dominance, in which warming, precipitation and year were treated as fixed factors, and block was treated as a random factor. Linear mixed-effects models were also used to assess the effects of warming, precipitation and their interactions on community biomass temporal stability, species asynchrony, the stability of the three different abundance groups (dominant, common and rare species) and the stability of the four functional groups (grasses, sedges, legumes and forbs), in which warming and precipitation were treated as fixed factors, and block was treated as a random factor. The significance threshold was pre-established as $\alpha = 0.05$.

Simple linear regressions was used to assess how species asynchrony, species richness, dominance, the stability of the three abundance groups and the stability of the four functional groups relate to community biomass temporal stability. Linear regressions were also used to assess the relationship between species richness/dominance and species asynchrony, and the relationship between species relative abundance and population-level stability. Common/rare species stability was natural log transformed to ensure normality. ANCOVA was used to test whether the slopes of the linear regressions differed significantly among the six treatments.

SEM was used to explore the pathways of how warming and altered precipitation, through influencing factors considered in the linear regressions, affected community biomass stability. We first considered a full model that included all possible pathways (Fig. 4 and Supplementary Table 3), and then sequentially eliminated nonsignificant pathways until we attained the final model (Supplementary Fig. 5). We used $\chi^2$ test, Akaike information criteria and the root mean square error of approximation to evaluate the fit of model.

All statistical analyses were conducted using R version 3.2.2 (R Foundation for Statistical Computing, Vienna, Austria, 2013), with the 'nlme' package for linear mixed-effects models, the 'vegan' package for calculating Simpson's dominance index, the 'ggplot2' package for plotting all histograms and regression figures and the 'lavaan' package for SEM[67].

**Data availability.** The data sets generated during the current study are available from the corresponding authors on reasonable request.

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

## Acknowledgements

We thank Li Lin, Fei Ren and Qian Chen for helping with field work and identifying plant species, and Xin Jing for helping with structural equation modeling. We also thank many workers of the Haibei Alpine Grassland Ecosystem Research Station for helping maintain our experiment. This study was supported by the National Basic Research Program of China (2014CB954000), the National Nature Science Foundation of China (31630009, 31321061 and 31361123001), the National Science Foundation of the USA (DEB-1257858 and DEB-1342754) and the 111 Project of China (Grant No. B14001).

## Author contributions

J.-S.H. designed the experiment; Z.Ma, H.L., Z.Mi, Z.Z., Y.W. and W.X. carried out the experiment; Z.Ma analysed the data; all the authors contributed to writing; L.J. and J.-S.H. guided data analyses and writing and finalized the manuscript.

## Additional information

**Competing interests:** The authors declare no competing financial interests.

