## [Peer Review File · Nature Communications]

Reviewers' comments:

Reviewer #1 (Remarks to the Author):

Review: Climate warming reduces community temporal stability

The manuscript by Ma et al. reports on an impressive field study in grassland on the Tibetan Plateau. In an orthogonal manipulation of temperature and precipitation, the authors followed plant community biomass and community composition over five and four years, respectively. They study the stability of plant biomass production and relate the treatment effects to variations in the dominance structure of the plant communities and the asynchrony among species. Structural equation modeling was used to explain treatment effects on stability. They find that warming reduces community stability by decreasing asynchrony among plants. Further, the population stability of dominant species explained a significant proportion of the variability on community stability. By contrast, sub-dominant species and plant species richness did not significantly contribute to stability. The authors conclude that future climate change may alter the stability of ecological communities and the stable provisioning of ecosystem services.

Overall, this is a very well prepared and important manuscript. All the methods seem to be adequate, and the text reads very well. The results are novel and the conclusions justified. Thus, I do not have any major suggestions how to improve the manuscript (see some minor comments below). However, I am not sure if the message of the manuscript is that novel that it justifies publication in Nature Communications. The conclusion that climate change will alter the stability of community functions through species' asynchrony and the role of dominant species is not new as also explained by the authors in lines 171-182 (and references; e.g., Xu et al. 2016 J Ecol). The novelty of the present study is that such effects are shown for warming. Also, the mechanisms presented here are based on previous studies, while again, for warming these relationships may be novel.

Minor comments:

- Move Fig. 1 to Supplement
- Did you observe shifts in plant community composition, i.e. in the relative dominance of different plant functional groups, in response to the treatments, and could those shifts explain additional parts of the variance?
- Is the comparison of dominant versus sub-dominant species' contributions fair as effects may be dominated by biomass effects?
- L. 27-31: long sentence; consider having two sentences.
- L. 43 (temporal stability): provide reference for statement
- L. 65: given that warming
- L. 71: we report
- L. 75: by alpine grassland
- L. 75: has experienced
- L. 77: and greater
- Results: provide percental differences for treatment effects
- L. 100: little effect
- L. 112: Linear regression
- L. 143: Fig. 6
- L. 144: failed sounds strange; rephrase
- L. 159: species responses?
- L. 205: identified weakened

Reviewer #2 (Remarks to the Author):

The paper by Ma et al. details the community factors that lead to the stability of annual net primary production exposed to warming and rainfall manipulation. The paper is very well written and engaging. The topic is of wide interest and the results are extremely interesting. The science

itself is well designed and largely very well analysed. Therefore, I think that this manuscript should be published in Nature Communications. However, I do believe that there are some aspects of the paper that require improvement prior to publication. Some are about the manuscript itself but I do have a criticism of one aspect of the analysis as detailed below. Therefore, I feel that the manuscript requires revision before it can be considered ready for publication. These revisions are not major and, in my opinion, the manuscript does not need a further round of review, as the required revisions can just be handled by an editor.

The major issue I have with the analysis is the concept of turn-over or stability of species, particularly of minor species. In this study, three 15 cm square quadrats were harvested in each plot. While this will give a reasonable indication of above ground biomass, it will not really adequately sample minor species. Therefore, it is likely that the minor species were present but undetected by the sampling. To talk about turn-over or stability of these minor species is therefore quite incorrect when using this method. I believe that the enormous scatter of points for species with low relative abundance on Fig. 4b is due to this sampling problem. In fact, I believe that the entire relationship between stability and abundance is an artefact and should be removed. This is because it is all about detectability and the two variables (species relative abundance and stability) are going to be directly dependent upon their detectability. Of course more abundant species are more stable because you are always going to sample them whereas the more minor species are going to be sampled only a small proportion of the time, since such a small proportion of the plot was actually sampled. Therefore, I honestly believe that the Figure 4b and the entire relationship between stability and species abundance should be removed from the manuscript altogether. Figure 4a is fine because it only involves the dominant species and shows how treatments affect their biomass.

Fortunately, this relationship is not key to the SEM or conclusions of the paper, so these don't need to be completely repeated.

The second major point I have is that I object to the term "community stability" being used to represent stability of ANPP. Every time I read "community stability" I was thinking about community composition rather than stability of biomass production. I think many ecologists would think the same. In fact, the term "community stability" has been used to mean many different things and is one of those unscientific, undefined terms that should be avoided. I urge the authors to replace "community stability" with what they really mean, which is stability of biomass production.

Lastly, the authors link regression relationships with causality in the discussion. The authors have shown clearly that the stability of dominant species and species asynchrony both have a strong correlation with stability of biomass production, but they have NOT demonstrated that one causes the other. However, the discussion clearly makes such statements. I would recommend slightly more cautious wording because it is possible that some underlying process is driving all the variables in a similar way, rather than one driving the other.

Minor comments.

Fig. 1 What are the letters (a,b,c) representing in Figure 1. They are not mentioned in the caption. Do the photos show different aspects of the experiment? Is there a photo of the warming treatment IR heaters?

Fig. I think the use of circles for unwarmed and triangles for warmed treatments is excellent, but the colour scheme is not intuitive. Why not use the same colours for the watering treatments. In other words, the watering control could be blue, with a blue circle for unwarmed and a blue triangle for warmed controls. Similarly, the precipitation reduction treatment could be brown for both warmed and unwarmed and the added precipitation treatment could be green. This would make the figure much easier to interpret and would mean that the treatment effects are more obvious.

Reviewer #3 (Remarks to the Author):

The author's conducted a well designed long-term study on how climate warming and altered precipitation (high, low) affect biomass stability in Tibet. Stability is the perhaps the key measure of how ecosystems will respond to changes in the climate, and this study found an interesting result - that warming leads to reduced species growth asynchrony, and this in turn leads to reduced stability. The stability of dominant species was also a strong predictor of biomass stability at the system level. The work will make a major contribution to the fields of ecosystem ecology and global change biology. I have some comments on how to improve the paper, based on line numbers.

L46, change to 'can have a positive'

L40-41. This paragraph needs to be reworked, as it jumps around and does not have a logical structure. Several mutually non-exclusive mechanisms (I agree that they are non-exclusive). The mechanisms are the sampling effect, the asynchrony effect, and the dominant species effect. Yet, the paragraph talks about stability changing or not due to climate changes. That is, the paragraph mixes up mechanisms (listed above) with results. The paragraph should be rearranged to have a more logical flow. All of the content is good though, it just needs to be reorganized.

L61. NPP instead of community productivity?

L71. Should be 'report on a'

L75. drop 'the'

L77. drop 'its'

L100. Should be little, not litter.

L171. Should be 'temperatures'

This paper should cite Wilsey et al. (2014) Ecology Letters for support of the dominant species leading to high stability. This paper is important because it shows that a loss of diversity due to exotic plant species DID NOT lead to loss in stability. The reason there was no loss is because the species that became dominant over time tended to have greater than average stability.

Figure 1b. I would call it species richness, and not community richness. It is the number of species and not the number of communities. The number of communities would be beta diversity, which was not considered here.

Responses to reviewers' comments:

Reviewers' comments:

Reviewer #1 (Remarks to the Author):

Review: Climate warming reduces community temporal stability

The manuscript by Ma et al. reports on an impressive field study in grassland on the Tibetan Plateau. In an orthogonal manipulation of temperature and precipitation, the authors followed plant community biomass and community composition over five and four years, respectively. They study the stability of plant biomass production and relate the treatment effects to variations in the dominance structure of the plant communities and the asynchrony among species. Structural equation modeling was used to explain treatment effects on stability. They find that warming reduces community stability by decreasing asynchrony among plants. Further, the population stability of dominant species explained a significant proportion of the variability on community stability. By contrast, sub-dominant species and plant species richness did not significantly contribute to stability. The authors conclude that future climate change may alter the stability of ecological communities and the stable provisioning of ecosystem services.

Overall, this is a very well prepared and important manuscript. All the methods seem to be adequate, and the text reads very well. The results are novel and the conclusions justified. Thus, I do not have any major suggestions how to improve the manuscript (see some minor comments below). However, I am not sure if the message of the manuscript is that novel that it justifies publication in Nature Communications. The conclusion that climate change will alter the stability of community functions through species' asynchrony and the role of dominant species is not new as also explained by the authors in lines 171-182 (and references; e.g., Xu et al. 2016 J Ecol). The novelty of the present study is that such effects are shown for warming. Also, the mechanisms presented here

are based on previous studies, while again, for warming these relationships may be novel.

Response: We agree with the reviewer that our study did not identify previously unknown mechanisms contributing to biomass temporal stability. Given mechanisms contributing to biomass temporal stability have been extensively studied, our study aimed to understand how climate changes influence stability and known mechanisms. As the reviewer pointed out, the novelty of our study lies in the finding that climate warming reduced biomass temporal stability via reducing species asynchrony, independent of precipitation regimes. To our knowledge, neither the pattern (that warming influences stability independent of the precipitation effect) nor the mechanism (that warming alters species asynchrony) has been reported previously. Our study is novel also because we are the first to study both warming and precipitation effects on temporal stability in a single experiment, and because our experiment represents a rare study of the stability of the alpine grassland in a climate-sensitive region that is often referred to as the Third Pole, which is undergoing rapid climate change. We have made these points clear in the revised manuscript (lines 153-164).

Minor comments:

- Move Fig. 1 to Supplement

Response: Done

- Did you observe shifts in plant community composition, i.e. in the relative dominance of different plant functional groups, in response to the treatments, and could those shifts explain additional parts of the variance?

Response: We did observe a shift in plant community composition in response to our experimental treatments. After classifying plants into four functional groups, including grasses, sedges, legumes and forbs, we found that climate warming increased legume

biomass and decreased sedge and forb biomass, and that increasing precipitation increased the biomass of grasses, sedges and forbs (Supplementary Fig. 6). Linear mixed-effects models showed that experimental treatments did not affect the stability of the four functional groups (Supplementary Table 4). Regressions showed that the stability of the grass functional group, but not other functional groups, was significantly associated with community biomass stability (Supplementary Fig. 7). However, adding grass stability into the SEM did not significantly improve the fit of the model ($R^2=0.77$ and 0.75 in the SEM model with and without including grass stability, respectively). We have provided this information in the revised manuscript [see the newly created subsection of functional group stability in the Result (lines 126-130), lines 136-139, and lines 148-150].

- Is the comparison of dominant versus sub-dominant species' contributions fair as effects may be dominated by biomass effects?

Response: In our study, dominant species contribute most to community biomass, but there are few dominant species; subordinate species contribute relatively little to community biomass, but there are many subordinate species. While the common perception is that dominant species may largely determine ecosystem functioning (i.e., Grime's mass ratio hypothesis¹), some studies have shown that the role of subordinate species for ecosystem functions cannot be ignored²⁻⁴. Our study compared the role of dominant species vs subordinate (common and rare) for ecological stability, and found support for the mass ratio hypothesis. The consideration of dominant species stability also allowed us to compare its role with that of species diversity in influencing biomass temporal stability.

- L. 27-31: long sentence; consider having two sentences.

Response: Done. See lines 27-31

- L. 43 (*temporal stability*): provide reference for statement

Response: Done.

- L. 65: *given that warming*

Response: Corrected.

- L. 71: *we report*

Response: Corrected.

- L. 75: *by alpine grassland*

Response: Corrected.

- L. 75: *has experienced*

Response: Corrected.

- L. 77: *and greater*

Response: Corrected.

- *Results:* provide percental differences for treatment effects

Response: Done.

- L. 100: *little effect*

Response: Corrected.

- L. 112: *Linear regression*

Response: Corrected.

- L. 143: *Fig. 6*

Response: Corrected.

- L. 144: *failed sounds strange; rephrase*

Response: Done. See lines 176.

- L. 159: *species responses?*

Response: Corrected

- L. 205: *identified weakened*

Response: Corrected.

Reviewer #2 (Remarks to the Author):

The paper by Ma et al. details the community factors that lead to the stability of annual net primary production exposed to warming and rainfall manipulation. The paper is very well written and engaging. The topic is of wide interest and the results are extremely interesting. The science itself is well designed and largely very well analysed. Therefore, I think that this manuscript should be published in Nature Communications. However, I do believe that there are some aspects of the paper that require improvement prior to publication. Some are about the manuscript itself but I do have a criticism of one aspect of the analysis as detailed below. Therefore, I feel that the manuscript requires revision before it can be considered ready for publication. These revisions are not major and, in my opinion, the manuscript does not need a further round of review, as the required revisions can just be handled by an editor.

The major issue I have with the analysis is the concept of turn-over or stability of species, particularly of minor species. In this study, three 15 cm square quadrats were harvested in each plot. While this will give a reasonable indication of above ground biomass, it will not really adequately sample minor species. Therefore, it is likely that the minor species were present but undetected by the sampling. To talk about turn-over or stability of these minor species is therefore quite incorrect when using this method. I believe that the enormous scatter of points for species with low relative abundance on Fig. 4b is due to this sampling problem. In fact, I believe that the entire relationship between stability and abundance is an artefact and should be removed. This is because it is all about detectability and the two variables (species relative abundance and stability) are going to be directly dependent upon their detectability. Of course more abundant species are more stable because you are always going to sample them whereas the more minor species are going to be sampled only a small proportion of the time, since such a small proportion of the plot was actually sampled. Therefore, I honestly believe that the Figure 4b and the entire relationship between stability and species abundance should be removed from the

manuscript altogether. Figure 4a is fine because it only involves the dominant species and shows how treatments affect their biomass.

Fortunately, this relationship is not key to the SEM or conclusions of the paper, so these don't need to be completely repeated.

Response: We thank the reviewer for this insightful comment. We agree that the low detectability of rare species may potentially complicate our analysis of the relationship between species abundance and stability. To find out if this is an issue, we run additional regressions of species relative abundance and stability using only the dominant and common species, for which detectability is not an issue. We found similar patterns as when all species (dominant, common, and rare species) were considered (Supplementary Fig.3), suggesting that the observed positive relationship between species abundance and stability is not an artifact associated with species detectability. We have made this point clear in the revised manuscript (lines 121-124). As pointed out by the reviewer, these results are not key to our main findings; we thus have relocated them to the supplementary material section (Supplementary Fig. 3).

The second major point I have is that I object to the term “community stability” being used to represent stability of ANPP. Every time I read “community stability” I was thinking about community composition rather than stability of biomass production. I think many ecologists would think the same. In fact, the term “community stability” has been used to mean many different things and is one of those unscientific, undefined terms that should be avoided. I urge the authors to replace “community stability” with what they really mean, which is stability of biomass production.

Response: We agree and have replaced the term with “temporal stability of community biomass” or “biomass temporal stability” in the revised manuscript.

Lastly, the authors link regression relationships with causality in the discussion. The authors have shown clearly that the stability of dominant species and species asynchrony

both have a strong correlation with stability of biomass production, but they have NOT demonstrated that one causes the other. However, the discussion clearly makes such statements. I would recommend slightly more cautious wording because it is possible that some underlying process is driving all the variables in a similar way, rather than one driving the other.

Response: We have toned down our statements on the causality of community biomass stability in the discussion. The revised discussion is now more consistent with the correlative nature of our analysis.

Minor comments.

Fig. 1 What are the letters (a,b,c) representing in Figure 1. They are not mentioned in the caption. Do the photos show different aspects of the experiment? Is there a photo of the warming treatment IR heaters?

Response: These photos show different aspects of the experiment. We have moved the figure to the Supplement (Supplementary Fig. 1; per reviewer 1 suggestion), added a photo of the IR heaters in the warming treatment, and explained what each photo was in the figure legend.

Fig. I think the use of circles for unwarmed and triangles for warmed treatments is excellent, but the colour scheme is not intuitive. Why not use the same colours for the watering treatments. In other words, the watering control could be blue, with a blue circle for unwarmed and a blue triangle for warmed controls. Similarly, the precipitation reduction treatment could be brown for both warmed and unwarmed and the added precipitation treatment could be green. This would make the figure much easier to interpret and would mean that the treatment effects are more obvious.

Response: Done.

Reviewer #3 (Remarks to the Author):

The author's conducted a well designed long-term study on how climate warming and altered precipitation (high, low) affect biomass stability in Tibet. Stability is the perhaps the key measure of how ecosystems will respond to changes in the climate, and this study found an interesting result - that warming leads to reduced species growth asynchrony, and this in turn leads to reduced stability. The stability of dominant species was also a strong predictor of biomass stability at the system level. The work will make a major contribution to the fields of ecosystem ecology and global change biology. I have some comments on how to improve the paper, based on line numbers.

L46, change to 'can have a positive'

Response: Done.

L40-41. This paragraph needs to be reworked, as it jumps around and does not have a logical structure. Several mutually non-exclusive mechanisms (I agree that they are non-exclusive). The mechanisms are the sampling effect, the asynchrony effect, and the dominant species effect. Yet, the paragraph talks about stability changing or not due to climate changes. That is, the paragraph mixes up mechanisms (listed above) with results. The paragraph should be rearranged to have a more logical flow. All of the content is good though, it just needs to be reorganized.

Response: Per the reviewer's suggestion, we have reorganized this paragraph by discussing stability mechanisms first before touching upon environmental changes affecting stability. See lines 40-54.

L61. NPP instead of community productivity?

Response: We now use "community biomass" throughout our manuscript.

L71. Should be 'report on a'

Response: Corrected.

L75. drop 'the'

Response: Corrected.

L77. drop 'its'

Response: Corrected.

L100. Should be little, not litter.

Response: Corrected.

L171. Should be 'temperatures'

Response: Corrected.

This paper should cite Wilsey et al. (2014) Ecology Letters for support of the dominant species leading to high stability. This paper is important because it shows that a loss of diversity due to exotic plant species DID NOT lead to loss in stability. The reason there was no loss is because the species that became dominant over time tended to have greater than average stability.

Response: We agree and have included this reference in the revised manuscript.

Figure 1b. I would call it species richness, and not community richness. It is the number of species and not the number of communities. The number of communities would be beta diversity, which was not considered here.

Response: We have replaced “community richness” with “species richness” in the revised manuscript.

Cited references

- 1 Grime, J. P. Benefits of plant diversity to ecosystems: immediate, filter and founder effects. *J. Ecol.* **86**, 902-910 (1998).
- 2 Lyons, K. G. & Schwartz, M. W. Rare species loss alters ecosystem function – invasion resistance. *Ecol. Lett.* **4**, 358-365 (2001).
- 3 Mariotte, P., Vandenberghe, C., Kardol, P., Hagedorn, F. & Buttler, A. Subordinate plant species enhance community resistance against drought in semi-natural grasslands. *J. Ecol.* **101**, 763-773 (2013).
- 4 Mariotte, P. Do subordinate species punch above their weight? Evidence from above- and below-ground. *New Phytol.* **203**, 16-21 (2014).

REVIEWERS' COMMENTS:

Reviewer #1 (Remarks to the Author):

Dear authors,
thank you for addressing my concerns - I do not have any additional recommendations.
Best wishes,
Nico Eisenhauer

Reviewer #2 (Remarks to the Author):

The original ms was very good and really only needed rather minor alterations. The detailed and thoughtful response to reviewers' comments has indicated that the authors have taken all of the reviewers' comments in the spirit in which they were intended. The revised ms is therefore better than the original and in my opinion both worthy of and ready for publication in Nature Communications.

Well done to the authors and I look forward to seeing the paper published!

Reviewer #3 (Remarks to the Author):

The author's did a great job with the revision. This paper makes a great contribution to this topic.